



# Grand challenges of Wind Energy Science – Meeting the needs
# and services of the power system
Mark O'Malley[1], Hannele Holttinen[2], Nicolaos Cutululis[3], Til Kristian Vrana[4], Jennifer King[5],
Vahan Gevorgian[5], Xiongfei Wang[6], Fatemeh Rajaei-Najafabadi[1], Andreas Hadjileonidas[1]
[1]Department of Electrical and Electronic Engineering, Imperial College, Exhibition Rd, South Kensington,
London SW7 2BX, United Kingdom
[2]Recognis Oy, Espoo, 02200, Finland
[3]DTU, Department of Wind and Energy Systems, Risø Frederiksborgvej 399 DK- 4000 Roskilde, Denmark
[4]SINTEF Energy Research, Sem Sælands vei 11, 7034 Trondheim, Norway
[5]NREL, Power Systems Engineering Center, 15013 Denver West Parkway Golden, CO 80401, United States
[6]Department of Electrical Engineering, KTH Royal Institute of Technology, Stockholm, 114 28, Sweden
*Correspondence to*: Mark O'Malley (m.omalley@imperial.ac.uk)
**Abstract:** The share of wind power in power systems is increasing dramatically and this is happening in parallel
with increased penetration of solar photovoltaics, storage, other inverter-based technologies, and electrification of
other sectors. Integrating all these technologies in a cost-effective manner while maintaining (or improving) power
system reliability is challenging and is driving radical changes to planning and operations paradigms.    Wind
power can maximise its long-term value to the power system by balancing the needs it imposes on the power
system with its contribution to addressing these needs with services. Research in wind power should be guided by
this balanced approach and by concentrating on its advantages over competitors. The research challenges within
the wind technology itself are many and varied with control and coordination internally being a focal point in
parallel with a strong need for coordination with research in other technologies such as storage, power electronics
and power systems that together are fundamental and potentially profound. This is all driven by the unchanging
nature of the fundamental objective of power systems - maintaining supply demand balance reliably at least cost.
## 1 Introduction
It is widely accepted that the near-term focus for rapid and substantial emissions reductions in the energy system
is the decarbonisation of electricity and the electrification of other sectors of the economy (IEA, 2022). Wind and
solar photovoltaic (PV[1]) compared to the alternatives, i.e., costs and maturity of nuclear and/or carbon capture
and storage, are dominating newly installed electricity capacity globally (EIA, 2022; IEA, 2022; Lazard, 2023;
REN21, 2023).  This dramatic increase in wind and solar PV has prompted a recent review of the status of these
two key technologies to determine their long-term research challenges (Haegel et al., 2019; Veers et al., 2019;
Veers et al., 2022; EAWE, 2023). These papers arose out of an International Energy Agency (IEA) initiative

---

[1] While this paper is focused on wind, solar PV has very similar characteristics and impacts on power systems and therefore they are dealt with together where appropriate.  Wind and solar PV are sometimes collectively referred to as variable renewable energy (VRE) resources, but this collective term is only used in the paper where appropriate as the focus is on wind.



(Dykes et al., 2019). They identified a common research challenge, grid integration[2], the all-important task of
ensuring that with increased penetration of wind and solar PV in power systems[3] the primary objective of
maintaining supply demand balance reliably and at least cost is met (O'Malley, 2011).
Increases in wind, solar PV, and electrification of other energy carriers (e.g., hydrogen) and sectors (e.g., industry)
are also driving a rapid increase in other complementary technologies. These include flexible demand, electric
vehicles, sector coupling which more broadly includes heat pumps, storage (batteries, power to heat, and power
to hydrogen etc.), as well as changes to the grid infrastructure including offshore grids and increased high voltage
direct current (HVDC) grids (National Academies of Sciences and Medicine, 2021; Pineda and Vannoorenberghe,
2023). Many of these technologies come with their own integration challenges and opportunities (Matevosyan et
al., 2021). All these simultaneous changes lead to a high dimensional situation where meeting the primary
objective is not a simple problem of "wind integration" but rather a multidimensional energy systems integration
challenge (O'Malley et al., 2016). Wind power in this rapidly changing environment needs to adapt competitively
where it can, by reducing any negative impacts and contributing positively to meeting the power system' primary
objective. That is, wind power can become part of the solution instead of being part of the problem.

Instantaneous penetration levels of 100 % wind have already been reached in some regions (e.g., Denmark) but
have not been reached on a country scale synchronous power system[4] (Söder et al., 2020).  At current rates of
deployment and considering the stated targets, instantaneous penetrations of 100 % wind in synchronous power
systems will be a common event in coming years with 100% wind energy penetrations being approached in the
coming decades, e.g., Ireland (Denholm et al., 2021; Ireland, 2021). Over the past few years there have been many
papers on the topic of 100 % renewables which typically do not delve into the power systems challenges and are
therefore of limited utility. The reader is cautiously directed to many papers on the topic, see for example (Breyer
et al., 2022).

---

[2] When the challenge is related to a specific technology sometimes the specific technology is specified i.e., wind integration.  Although wind is the focus here, we will not use the term as one of the key messages is that wind or any technology cannot be treated in isolation from an integration perspective because there are so many changes happening simultaneously.

[3] There is a plethora of terminology that can be confusing, including grid, power grid, network, power system, electricity system etc.  Without defining them strictly the grid, power grid and network are all similar and refer to the network of transmission and distribution lines and associated equipment i.e., the "wires".  The power system and electricity system are similar, power system being the more colloquial in the engineering community, and includes in addition to the grid the generation and demand etc.  Therefore, the grid and wind are part of the power system.  However, grid has been adopted as the term of common use even when power system may strictly be more correct.  Here both terms grid and power system are used throughout with best endeavours to avoid ambiguity.

[4] Virtually all power systems are synchronous, but it is not necessarily defined by a geographical, political and/or commercial boundary but by a technical boundary. Denmark's power system is part of two synchronous power systems one in the west (part of the much larger European Continental synchronous area) and one in the east (part of the larger Nordic synchronous area). Ireland and Great Britain have their own synchronous power system.





Maintaining supply demand balance reliably and at least cost in high penetrations of wind (and solar PV) comes
with significant integration challenges (Hodge et al., 2020; Holttinen et al., 2020; Denholm et al., 2021). To
describe the opportunities for wind power in this integration challenge, a systematic framework that coincides
with planning, operational and market practices is required to define a set of power system needs and power
system services (Chaudhuri et al., 2023). Historically the needs and services paradigm were inherently embedded
into the vertically integrated utility concept and was not explicitly stated and more recently it formed the basis of
electricity markets (Schweppe et al., 2013; Kirschen and Strbac, 2018). Most fossil fuel, nuclear and hydro
generation are interfaced to the grid using synchronous machines (SMs)[5] and are the main providers of most of
the power system services. When these SMs are in synchronism with one another, they form a synchronous power
system.  SMs are well known and understood for decades, and they are at the heart of power systems, their
planning, and operations (Glover et al., 2012). Increased wind power and the other technologies are changing
existing power systems and therefore the needs and if they replace other technologies e.g. SMs there is a reduction
in the supply of some services.  This requires wind power, and all other technologies connected to the grid to meet
these changing needs with services (EirGrid, 2023).
**1.1 Impact of wind power on power systems**
From an integration perspective the characteristics of wind and solar PV (ESIG, 2019; Denholm et al., 2021) fall
into two broad areas:
(1) the operational variability, uncertainty, and distributed nature of the primary energy source that impacts the
economics of the power system but also on the reliability as penetration levels increase
(2) the interface of most of the wind and solar PV to the grid is not by SMs but by power electronic
inverters/converters[6] that are highly controllable, non-synchronous and have limited overloading capabilities that
mainly impact on the reliability of the power system.

The operational variability, uncertainty, and distributed nature of the primary energy resource is a major difference
of wind and solar PV when compared to other primary sources such as fossil fuel, nuclear generation, and hydro
with pondage (Bird et al., 2013). It is not a case that these other primary sources are not variable, uncertain, and
distributed, they are, but the characteristics are different. These differences are all related to the fact that these
primary energy sources can all be easily stored, hence the variability and uncertainty can be buffered.  Fossil and
nuclear fuel can also be transported to take advantage of economies of scale at a centralised location. In the case
of wind and solar PV the primary energy source cannot be stored and hence the generation is more distributed,
and the full variability and uncertainty needs to be balanced internally by wind and solar PV by curtailing or by
other parts of the power system such as other generation, demand, or storage which are described in the literature
as the need for flexibility (Lannoye et al., 2012). Curtailing wind and solar PV (or storing it) need the right

---

[5] Most SMs in capacity terms in power systems are synchronous generators but other SMs include synchronous
condensers and synchronous motors that have very similar characteristics from a services perspective.
[6] Converters go from alternating current (AC) to direct current (DC) and inverters go from DC to AC. Modern
wind generation for technical reasons produces AC that is then converted to DC and is then inverted to AC that is
injected into the AC grid. Inverter and converter are sometimes used interchangeably.



economic incentives and can therefore be part of the solution providing flexibility instead of solely demanding
flexibility from the system (Morales-España et al., 2021).

In the early days of the wind industry (1980s to 2000s), from a reliability point of view the "do-no-harm"
philosophy was adopted, i.e., if there were problems on the power system the wind power would invariably just
disconnect so as not to cause any further reliability issues (Christensen, 2010; Lauby et al., 2011; Zavadil et al.,
2011). Therefore, wind power did not impose any additional needs on the power system and the only service wind
provided was energy that was dependent on the wind availability on the day, with little or no operational planning
to account for wind variability and uncertainty with a forecast. At that time, driving down Levelized Cost of
Energy (LCOE) was the major objective of the wind industry as it still needed heavy subsidies.  This modality of
operation was pervasive until around one/two decades ago when wind power started to represent a significant
portion of energy provision on some power systems. Inevitably some needs driven by wind were recognised, and
additionally some services were required and/or incentivised from wind power such as forecasting, fault-ride-
through, active power regulation for frequency control, and local voltage control that had been adopted in several
power systems (Mohseni and Islam, 2012; EirGrid, 2019; IRENA, 2022).

During this period the concept of calculating the so-called integration costs was mooted (Smith et al., 2007;
Ueckerdt et al., 2013). There is a wide literature in this area even though it can be easily shown it is impossible to
calculate in an absolute sense the integration cost of a specific technology e.g., wind, and remains a controversial
subject (Müller et al., 2018). Relative integration cost can be calculated by comparing two different portfolios,
i.e., two power systems with different portfolios of generation and other technologies that serve the same demand
with equal reliability can have their costs compared (EirGrid, 2008; Holttinen et al., 2019). The portfolio cost is
highly system dependent and can increase rapidly as penetration levels of wind increase (Cochran et al., 2014).
For example, in a power system that is inherently flexible the cost of operational variability and uncertainty, while
maintaining reliability may be negligible up to a point where the flexibility saturates, portfolio costs can then rise
rapidly (Figure 1). These costs can and do occur throughout the power system and can range from being directly
part of the cost of the wind or costs elsewhere in the power system, but as stated above it is impossible to allocate
these costs to a specific technology.

**1.2 Wind technology for power system integration**

There exist four distinct wind turbine generator technology classifications (Figure 2).  Type I & II dominated the
early years but are no longer deployed at scale.   Power electronics began appearing in wind turbines around 20
years ago with the development of   Type III wind turbines that employs a wound rotor induction generator with
a four-quadrant power converter. This addition affords the capacity to regulate rotor circuit currents, resulting in
an expanded operational range compared to Type I & II. Type III is also labelled as Doubly Fed Induction
Generator (DFIG), is characterised by flux-vector control, effectively decoupling active and reactive power
components to optimise output power.  DFIGs have many advantages over Type I & II



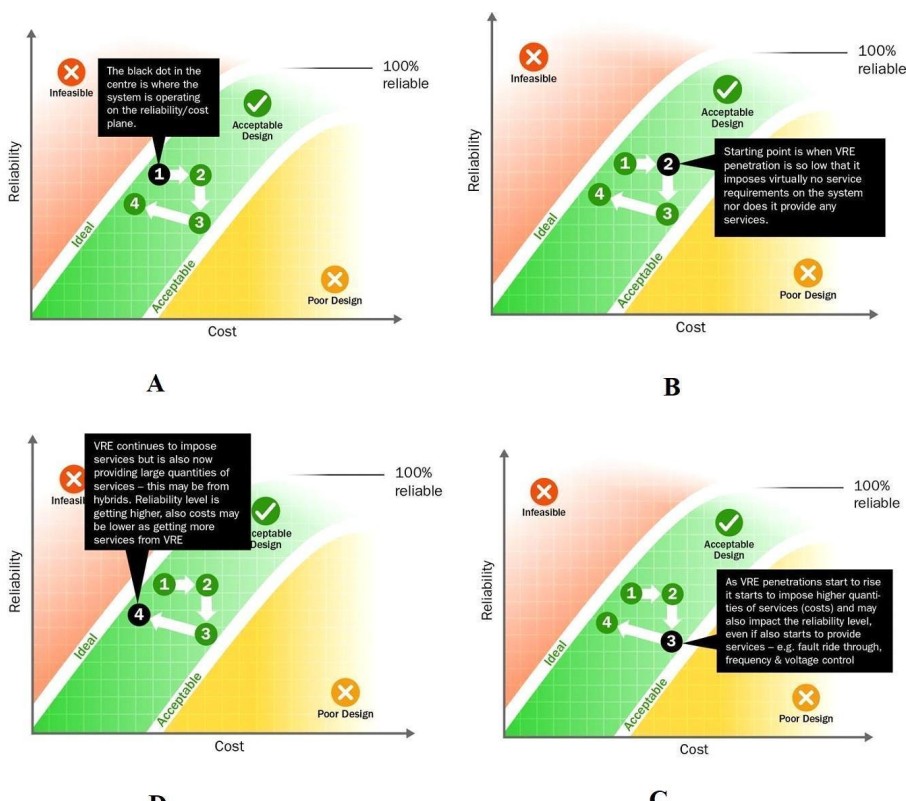

**Figure 1: Cost-reliability plane for power system design.** The ideal design boundary is inherently maximising the benefit by minimising the cost subject to a required level of reliability. Above this ideal there is an infeasible region and there is also an acceptable design boundary below which the power system design is deemed to be poor and not acceptable. Between the ideal and acceptable boundaries is the acceptable design region where the design is deemed to meet the primary objective. Impact of variable renewable energy (VREs) resources, wind and/or solar PV may deteriorate the reliability and may incur some additional costs. The progression from A through D represents a possible trajectory that is reflective of the evolution of designs as VRE penetrations increase and as for example services from VREs evolve dependent on research.

including lower cost, lower weights, and the ability to control both active and reactive power (Ackermann, 2012). Type IV involves either a synchronous or induction generator, connected to a full-scale back-to-back frequency converter isolates electrical generator dynamics from the grid (Singh and Santoso, 2011). Type IV is now the dominant wind turbine technology and has additional advantages such as optimised operation over all wind speeds, full control of active and reactive power production, augmented transients handling capability and improved power quality (Chen et al., 2009).

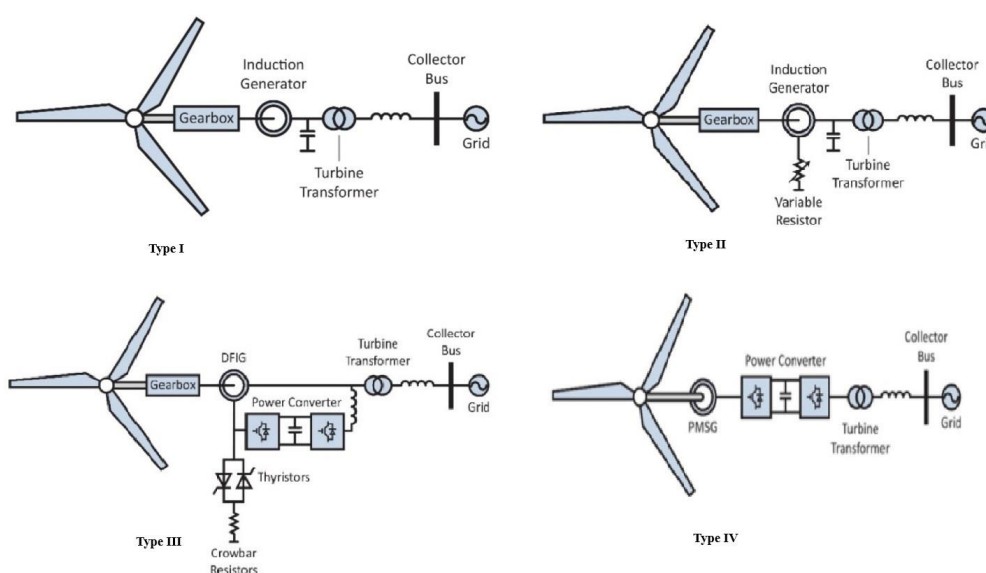

**Figure 2: The four types of wind turbine generator technology. Type I: squirrel cage induction generator model. Type II: squirrel cage wound rotor induction generator with external rotor resistance model. Type III: double-fed nonsynchronous generator model. Type IV: full power converter generator model ((Osman et al., 2018), used with permission).**

The power electronic inverters/converters that connect wind and solar PV to the grid have fundamentally different characteristics than SMs with respect to their ability to, e.g., maintain the frequency of the power system within stable limits. Other new technologies that are similarly interfaced to the grid by power electronics may add to these challenges include batteries, HVDC etc. Inverter Based Resources (IBR) is used to collectively describe these technologies that are interfaced to the grid by power electronic inverters/converters (Lin et al., 2022). The penetration levels of wind and solar PV are now so high in many power systems that they are beginning to replace fossil fuel generation (and even nuclear in some cases) in planning and operational time frames (Chaudhuri et al., 2023). Therefore, there is a trend towards the disappearance of SMs and their replacement with IBRs. This trend has consequences: a change in the needs, and the supply of services, which may in the future be supplied by IBRs including wind. Power systems that will approach 100% renewable generation will face the dilemma of keeping SMs with significant cost implications to provide services or relying on wind and/or other IBR-based technologies (solar PV, battery etc.) to provide the services (Hodge et al., 2020). Simply keeping SMs is not in itself a perfect solution since they may not be at the correct location or may be too expensive (Appleby and Rositano, 2019).

Of note is the ability, via the controls, of IBRs to not just operate in grid following mode (that is, other technologies, typically SMs in the power system setting the frequency and voltage for them to follow), but also the ability to be grid forming (capability to set and maintain frequency and voltage) and can also combine





characteristics of both (Kroposki et al., 2017). This is an extremely active research field in the power systems and
power electronics domains (Ackermann et al., 2017; Kroposki et al., 2017). For example, there is significant
research activity to try and establish what is the best ratio of grid forming and grid following inverters in an IBR-
dominated system and the current estimates have the grid forming penetration at around 20 ~ 30% (Matevosyan
et al., 2021). While the fundamental principles of IBR controls are a research topic regardless of the host
technology, the implementation in wind turbines will need to be adjusted or tailored to their characteristics (Veers
et al., 2023). Similarly for other host technologies, power-electronics-related solutions will be developed for IBRs
such as solar PV and battery storage, and this blurring of the boundaries between the technologies is to be expected
as the power system becomes much more integrated, and the wind technologies become more heterogeneous.

Another example of this heterogeneity are the research challenges for planning and operating offshore grids
(Cutululis et al., 2021; Tande et al., 2022). These include optimising the stepwise offshore grid buildout including
offshore energy hubs and hybrid AC/DC grids, considering uncertainties and the long lifetime of the
infrastructure, future amount of connected wind capacity and hydrogen demand. Other additional considerations
are dynamic electrical cables for floating wind power plants, and either floating or subsea substations to connect
the wind power plant to the offshore transmission grid, as well as subsea collection systems for grid connection
of large floating wind power plants.

Hybrids are another good example of heterogeneity e.g. wind power combined with storage and solar PV which
can help to lower the impact of variability and uncertainty and hence reduce the need for some services (Stenclik
et al., 2022). In hybrid plants for example the addition of storage, across timescales from short duration to even
long duration storage in future, and the addition of solar PV, which may be a complementary resource to wind,
will help increase the plant's ability to provide services to meet the needs of the power system (Nema et al., 2009;
Stenclik et al., 2022). It is important to note that the advantages of combining the technologies come from shared
transmission capacity, quantity of power electronics and controls which can therefore, at a reduced cost, maintain
the performance and the overall quality of the services provided to the power system. There are also regulatory,
subsidy, commercial and market advantages and can be very system specific. There are no purely synergistic
technical advantages inherent in hybrids and from a services perspective there may be limitations imposed by
constraining the technologies to act in a coordinated manner (Stenclik et al., 2022; Kemp et al., 2023).
**1.3 Future needs and services an opportunity for wind power**
For the power system to meet its primary objective wind and other technologies need to either adapt to the power
system and/or the rest of the power system needs to adapt to them in much the same way as has occurred
historically with SMs (Figure 3). Wind no longer must be accommodated but rather the power system needs to
enable its increased penetration. The end destination in this process is far from clear but will be heavily influenced
by research and innovation in power systems and the technologies that make up the power system including wind
(Veers et al., 2019). In this changing environment, power system needs and services should be assessed to ensure
power system reliability at least cost and/or to avoid providing services that are no longer needed which can be
costly. Resources, policy environment and stage of development are different across the world and there is and



will be a wide variety of power systems with distinct characteristics and while most power system needs and
services will share a lot in common some of them will be system specific.


**Figure 3: Wind adapting to the power system and the power system adapting to wind - a pathway to a cost-effective reliable power system.**


The required services can be found from different parts of the power system but here we focus on wind-based
solutions (Figure 4) that require significant research in the coming decades and that may have the potential to be
competitive against other sources of these services. For wind this translates into a multiscale research and design
challenge at the individual turbine, plant, hybrid level, with mechanical, electrically and/or control centric
solutions to the provision of services and a technical/economic comparison with other alternative sources for these
services to meet the power system needs (Figure 4). For other technologies such as solar PV and batteries this
translates similarly.



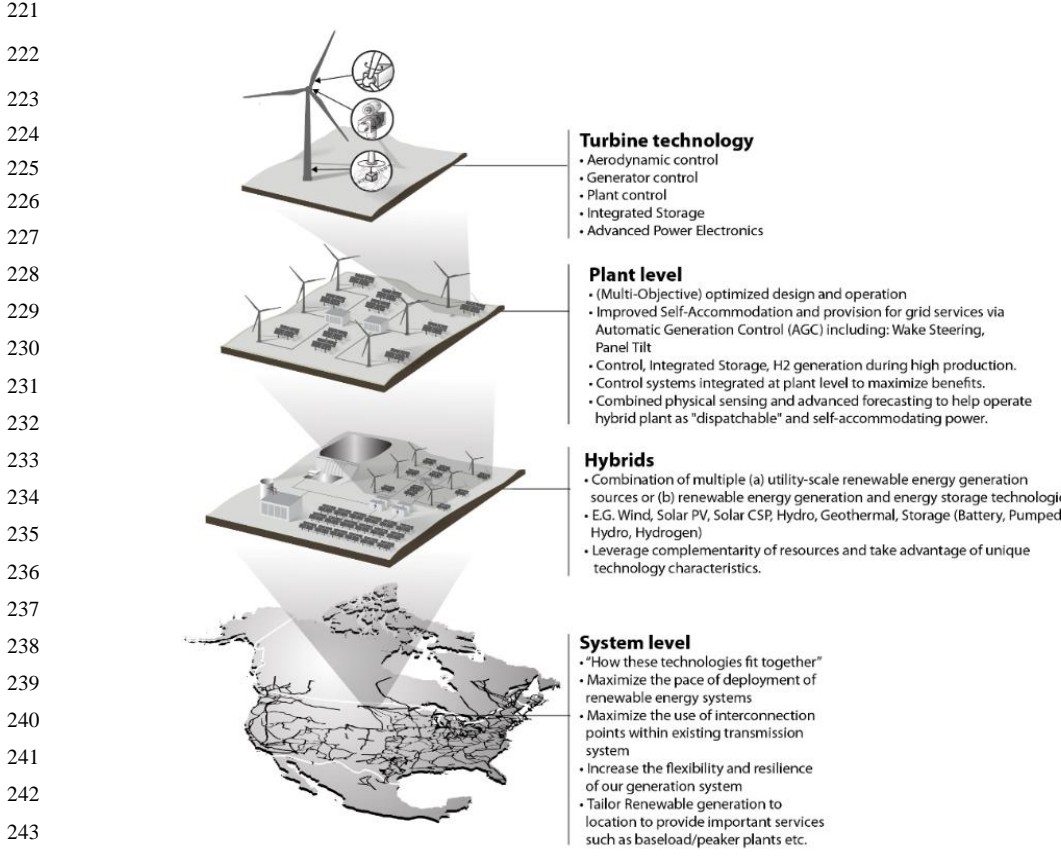

**Figure 4: Wind turbine to plant to hybrid to power system level and the provision of services to meet the power system needs.**

Procurement of services to meet the needs can range from a mandated capability (i.e. interconnection requirements/grid codes) to a capability that is paid for in a monopoly or a formal competitive market scenario and all the variations in between. In some cases, the need may be inherently met with no scarcity and hence no need for a mandate or incentives (Ela et al., 2019; Ela et al., 2021). This variation is evident in the range of services that exist in many of the electricity markets across the world (Rebours et al., 2007a, b). Procurement mechanisms are evolving in parallel to incentivise the services to meet the needs in an optimal manner (Hobbs et al., 2022). It is important that the wind industry is incentivised, by market signals, grid codes etc., through design and innovation to mitigating needs and/or providing competitive power system services. The wind industry needs the right incentives, so wind technology evolves to maximise its value to the system rather than solely maximising its power output. With wind and solar PV having near zero marginal cost this is also challenging electricity market and policy design (Neuhoff et al., 2023). Innovations in response to wind such as societal acceptance and lifestyle



adaptations to energy availability are also changing (Schuitema et al., 2018; Steg et al., 2018). These non-technical
challenges are not addressed in this paper.

There is no ideal way of cleanly defining needs and services as the power system is a highly integrated continuum
of overlapping, interacting technical characteristics from a wide range of technologies all acting in unison to
maintain supply demand balance reliably and at the lowest cost. Needs are not necessarily met by individual
services, but by combinations, while other characteristics and functionalities enhance a service that meets a need
or reduces a need. Furthermore, the needs themselves are uncertain into the future as they are a subject of the
evolving research in the power systems community, the technical characteristics of the IBRs and other new
technologies, and the reliability requirements which are also evolving as are technological methods that can
actively maintain security standards (Hedman et al., 2011; Bialek et al., 2021; O'Malley, 2022). Here we adopt
the work of the Global Power System Transformation Consortium (G-PST), and their structure that highlights
eight distinct needs: energy & capacity, and the six technical needs further grouped into frequency & voltage
control, synchronisation and damping, and protection and restoration (Figure 5).

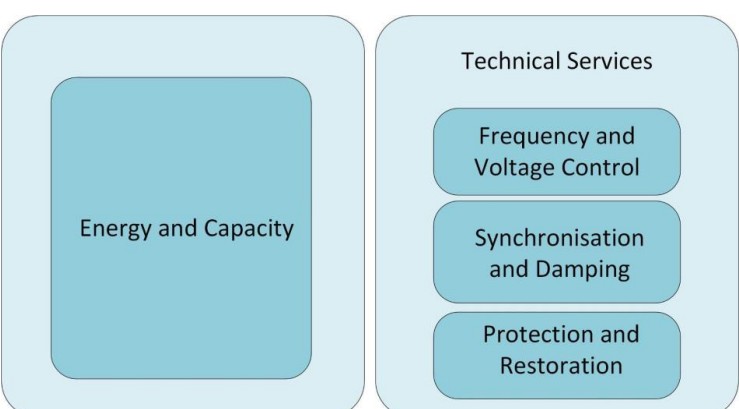

**Figure 5:  Power system services adapted from (Bialek et al., 2021).**

In the following four sections wind technology is assessed with respect to meeting these needs in electricity
systems driven by increasing penetration of wind and other technologies. This is achieved in the context of
changing power system needs and the removal of technologies that provide the services to meet these needs, with
the main example being the potential decline of SMs. Section 2 considers Energy and Capacity, Section 3
Frequency and Voltage Control, Section 4 Synchronisation and Damping and Section 5 Protection and
Restoration. Section 6 has a Discussion and Conclusions.



**2. Energy and Capacity**
Energy and capacity are the two most basic power system needs as supply demand balance cannot be met if there
is not enough energy, or if at any time instance or location there is not enough available capacity (generation,
transmission grid, etc.). In many power systems, wind is providing a significant proportion of the energy needs.
For example, in Denmark wind accounts for over 50 % of the electricity demand on average, more than 100% in
some instances (Holttinen, 2023), and several jurisdictions are targeting 80 to 100 % annual energy from wind
and solar PV, with wind being dominant in some cases such as in Ireland (SEAI, 2023). In many respects the
energy need/service is the primary focus of the other papers in this Grand Challenges series which focus on
lowering their costs and increasing their lifetime, reliability, and performance (Veers et al., 2022; Veers et al.,
292 2023).


Resource adequacy, the ability to meet the demand, has traditionally been about capacity and is equivalent to
having the generating capacity and transmission to serve the demand at every point in time and at every location
(Schweppe et al., 2013). It could traditionally be approximated as the ability to meet peak demand, but wind and
solar PV is changing this and moving the critical time to other periods of low wind and/or solar PV e.g. peak
aggregate net demand (demand less wind and solar PV). This is undermining some of the calculation
methodologies that were based on assumptions that may no longer be valid and may need to evolve (Stenclik et
al., 2021). Having abundant volumes of energy at times when demand from the consumer is low is of minimal
value and vice versa. Therefore, when the timing of energy from a technology correlates with demand, this
increases the capacity value of a technology and is fundamental to maintaining the reliability of the power system
(Keane et al., 2011). SMs with readily storable primary energy sources (e.g., fossil fuels, nuclear and hydro with
pondage[7]) are dispatchable and can therefore be available at peak times and have capacity values close to the
maximum of unity (the minimum is zero). However, the capacity value of wind (and solar PV) is relatively low
compared to SMs, ranging between 10-35% (Denholm et al., 2019; Holttinen et al., 2021).

The capacity value of wind reduces with wind penetration due to the correlation effect and it varies from year to
year depending on weather patterns (Hasche et al., 2010; Cradden et al., 2017). Therefore, with increasing wind
(and solar PV) penetration and the displacement of SMs the contribution of wind declines leading to an adequacy
deficit, which economically constitutes the single biggest challenge to very high penetration of wind and solar PV
(ESIG, 2019). Any design of wind power technology that improves the generation at lower wind speeds and
increases the number of running hours should improve the capacity value of wind. For example, low wind designs
with larger rotors may in the correct circumstances increase its capacity value (Dalla Riva et al., 2017; Wiser et
al., 2020; Swisher et al., 2022). In general, for best impact on capacity value, diversification of turbine types is
key as it reduces the correlation between their energy output, for example transmission capacity enables this
diversification as does mixing of onshore and offshore wind (EPRI, 2022a). An extreme version of this and a
long-term research topic is airborne wind power plants that can harness the wind resource at a high altitude that

---

[7] These primary energy sources can also be variable due to e.g. gas supply limitations, drought etc. This broadening out of what can impact resource adequacy is part of the evolution of the methodologies.





is steadier than regular wind turbines with potentially far higher capacity values but face very significant research
challenges (Kolar et al., 2013; Cherubini et al., 2015; Bechtle et al., 2019).

As stated above, needs and services are overlapping and interacting. A good example of that is energy and capacity
as is evidenced in the debate around energy-only markets (where capacity is incentivised by high energy prices at
periods of scarcity) and capacity markets (where capacity is directly rewarded) (Hogan, 2005). The focus is now
shifting from minimising LCOE towards maximising the value of wind, which effectively captures all the services
(Denny and O'Malley, 2007; Dykes et al., 2020; Loth et al., 2022). It no longer matters that the cheapest possible
electrons are being put into the grid (as energy), but what matters is when and where those electrons are put into
the grid (requiring the energy and capacity) adding additional value to the system.  Typically, energy and capacity
are the most valuable services but there are indications that as we approach higher penetrations of wind and solar
PV the more technical services, covered below and/or developed to meet future needs, could see their relative
value increase (Ela et al., 2017). Therefore, with the changing needs and services there is a need to strike a balance
between maximising the value of energy and capacity without potentially undermining the cost effectiveness or
ability of providing the technical services which could unduly increase the portfolio cost and/or degrade the
reliability of the power system (Figure 1). A good example of this can be found in this series of Grand Challenges
papers where plant controls can increase the energy yield of wind power plants, reducing wake losses and loading
of components and are also the way that wind can provide more cost-effective power system services (Meyers et
al., 2022; Veers et al., 2023). Energy management systems that seamlessly provide energy and other services
optimally are key to provide profitability of wind plants to owner operators while maximising value to the power
system when needed (Van Dijk et al., 2017). This is an ongoing research topic (IEA, 2023). These energy
management systems will need forecasting to determine the expected energy and other power system services
from wind and to do so in a coordinated manner, e.g., if wind is curtailed the wind capability after curtailment
needs to be estimated (Göçmen et al., 2018).

The use of wind-based hybrids together with solar resources (the less correlated with wind the better) can
maximise grid capacity and at the same time reduce the need to build out the grid and be beneficial.  This approach
can result in higher aggregated capacity factors[8] (average power output with respect to maximum rated output),
even if the shared point of interconnection can reduce the capacity factor of each individual technology (EPRI,
2022b).  Hybrids with storage can make a significant contribution to the adequacy need and is an active area of
research and deployment (Murphy et al., 2021). The storage device can be used across timescales to move the
energy from less into more valuable time periods. This is true regardless of whether the storage device is co-
located with the wind or not; however, directly coupling wind with solar PV and storage can bring down costs,
especially when network constraints exist (Jorgenson et al., 2018; Mallapragada et al., 2020). Non-hybrid
solutions where wind and storage are not subject to the same constraint may be more beneficial (EPRI, 2022b).

---

[8] Not to be confused with capacity value, defined above.  Capacity factor is a metric that is more related to
minimising the LCOE whereas capacity value is directed towards maximising value.



The continuing development of power system coupling to other energy sectors gives rise to new opportunities
for using the energy generated from wind (Van Nuffel et al., 2018). Power-to-X solutions, producing hydrogen
and its derivatives, e-fuels like ammonium and methanol, synthetic gases, may in future occur locally at wind
power plants, if the grid is congested, and providing storage for wind (Singlitico et al., 2020).

Future weather dependent power and energy systems require more data and improved tools, in which wind energy
needs to be well represented. Wind and solar PV impacts on power systems give rise to consideration of energy
adequacy (not enough energy resources for generation even if installed capacity is adequate) and the adequacy of
other services (EPRI, 2022b). In the future, as not only power generation but also demand will be increasingly
weather dependent due to electric heating and cooling, correlated events caused by common weather patterns need
to be considered more carefully when determining resource adequacy at the planning of energy systems
(Novacheck et al., 2021; Stenclik et al., 2021). Different weather authorities are working on improving seasonal
weather forecasts. The focus of their work lies on calculating the forecast uncertainty using probabilistic ensemble
forecasts instead of improving the expectation value of the forecast (Leutbecher and Palmer, 2008). Seasonal
demand shifting of industrial loads also has potential but is nascent in its research and development (Yang et al.,
370  2020).


Research to drive down LCOE is now being overtaken by the need to maximise the value of the wind with respect
to evolving energy and capacity needs and other services, which are themselves evolving.  To achieve this, the
research focus is being pushed outside the wind technology and towards storage technologies and e.g. Power- to-
X that needs to work in a coordinated manner with the core wind technology.  The balance between these becomes
a higher dimensional co-optimisation problem, in which all potential value streams are accounted for including
the six more technical needs/services of the power system. These six technical needs/services focus primarily on
the reliability of the power system starting with the power system's two most important attributes, frequency, and
voltage.

## 3. Frequency and Voltage control

The primary technical attributes of a power system are its frequency and voltage.  The frequency is a global
attribute that should be kept within preferred limits around a nominal value (typically 50 or 60Hz) throughout the
grid. The voltage is a local attribute that should be kept close to constant, which is different in different locations
(ranging from a few hundred volts to just over one million volts).  Controlling the frequency and voltage around
their nominal values is critical to maintain reliability of the power system.

### 3.1 Frequency control

Frequency control directly addresses flexibility. Too little supply or too much demand reduces frequency and
requires an increase in generation or a decrease in demand; too much supply or too little demand increases the
frequency and requires a decrease in generation or increase in demand to maintain frequency within its preferred
limits. Frequency control will be impacted both by variability and uncertainty of aggregate net demand as well as



by the relative number of SMs and IBRs online. There is considerable beneficial smoothing of variability and
predictability (in normalised metrics), but increased wind and solar PV will result in a more variable and less
predictable net demand. This drives the need for frequency control and more flexibility resources throughout the
power system. The power electronics and most importantly its control that interfaces wind to the grid typically
decouples the mechanical inertia of the wind turbine from the grid. Inertial response from SMs is inherent,
instantaneous, and impossible to prevent and with large numbers of SMs (mainly synchronous generators) on a
power system traditionally it was abundant and there was no need to reward it (Eto et al., 2010; Muljadi et al.,
2012). If SMs are replaced by IBRs this will lead to a reduced inertial response and the frequency control need
may not be met, and alternatives need to be found (Doherty et al., 2005; Ela et al., 2013; NGESO, 2023). This
inertia issue has had significant research attention recently although in some smaller systems it has been central
for several decades, emphasising the point that every system is different (Mullane and O'Malley, 2005; Doherty
et al., 2005; Mullane and O'Malley, 2006; Denholm et al., 2020).

The time scales of variability & predictability and inertial response are different. The variability & predictability
results in a need for relatively slow frequency control while the inertia issue results in the need for faster frequency
control. Frequency control goes from very fast "inertia" like response times (seconds and below), to fast frequency
response (seconds), to "primary frequency response" or "governor response" (seconds to minutes), to "regulation
reserve" or "automatic generation control reserve" to slower "flexibility reserve" or "load following" (minutes
and beyond) – eventually frequency control merges with the energy service (Ela et al., 2019). This highlights that
these needs lie on a continuum and the overlapping nature of the services as for example faster frequency response
reduces the need for inertial response (Delille et al., 2012).

Wind turbines can provide frequency control services and have done so for at least a decade (Ela et al., 2014).
Wind turbines can be curtailed to decrease generation to decrease frequency and when curtailed wind can increase
generation to increase frequency. Typically wind turbines will control their pitch and torque to change their power
output. These control mechanisms operate in the order of seconds, something that allows them to follow reference
signals that operate in 4 second intervals for example, which is the time scale of automatic generator control in
many power systems (Bevrani and Hiyama, 2011). Therefore, wind can provide many aspects of frequency control
services but may not be competitive because of the energy losses due to curtailment of wind. In some operational
circumstances such as stability constraints and minimum generation constraints, wind may already be partly
curtailed making frequency control more competitive (Denholm et al., 2019). One way of avoiding the curtailment
of wind when providing frequency control in an upward direction is to use the energy stored in the rotating blades
that can provide frequency control by slowing down the blades, without any pre-curtailment of wind energy.
While this effect may only be temporary (as there is a limit to how much you can slow down the blades), it can
work together with other mechanisms to help restore frequency to a stable level (Bonfiglio et al., 2018). Wind as
part of a hybrid with storage can also provide frequency control services. One of the challenges to be addressed
is the optimised control and operation of wind-based hybrids for frequency control support (Liu et al., 2019).




The fast controls of wind power plants (operation in seconds) have proved to give good compliance support in
low inertia operation (Ela et al., 2014; Denholm et al., 2021). For faster responses, some open issues remain, like
the ability of measuring, computing, and transmitting the frequency signal to the wind turbine/plant controller fast
enough, with a response time in the 100s of milliseconds scale. Operating wind turbines power electronics in grid
forming mode would significantly decrease their reaction time and, to some extent, mitigate the challenge with
measuring, computing, and transmitting the frequency change. The enhanced static synchronous compensator (E-
STATCOM) that integrates supercapacitor and grid forming control emerges as promising technology for
furnishing wind farms with inertial power response (Zhao et al., 2023).
While initial field tests show the viability of such operation, they also highlight challenges, with the most
pronounced one being the limited energy stored in the wind turbine rotor which if not properly managed, can lead
to opposite effects, i.e., reducing the power infeed to zero (Roscoe et al., 2021). Also, grid forming operation of
wind turbines will most likely have an impact on the drivetrain, and the known mechanical impacts on wind
turbine drive trains that need to be addressed and can result in additional costs and reliability impacts that may
not be justified (Girsang et al., 2014; Gloe et al., 2021; Zhang et al., 2021; Chen et al., 2022; Nguyen et al., 2022).
Under certain grid disturbances, such as phase angle jumps, active power oscillations may be induced within grid
forming wind turbines, which tend to propagate to the drivetrain of wind turbines and even trigger torsional
vibrations, degrading the lifetime and reliability of mechanical components (Avazov et al., 2022; Lu et al., 2022).
The recent work in (Roscoe et al., 2020) has shown that without additional energy storage, grid forming wind
turbines have limited power responses under some unscheduled frequency disturbances of power grids. This
highlights that the provision of services can have an impact on the cost and reliability of the wind technology and
its performance. Another relevant example is when a turbine provides frequency control and changes its pitch
and/or torque, which has implications on their wakes that propagate downstream and impact downstream turbines.
These impacts can be modelled using wake models and can affect the performance of downstream turbines if not
taken into account properly (Houck, 2022; Meyers et al., 2022). While most studies assume that wind turbines are
capable of accurately providing a short-term increase in their active power production, they usually employ very
simplified representations of the flow, e.g., turbulence is not represented, even if it can have a large impact on the
ability of the wind turbine in providing that extra active power (Veers et al., 2023). The energy contained in the
air flow field generally supports the provision of the desired short-term increase, as a power increase from
curtailed state to above available power is possible (until the resulting wake propagates to the adjacent turbines),
but the implications for service provision have not yet been investigated in detail. Plant control and flow control
can also take advantage of wake steering to distribute curtailments smarter and reduce losses when providing
services. Wake steering happens in the time scale of minutes, so this is not as useful for the fast response services.
An important part of frequency control is the ability to maintain the desired level of response in accordance with
the service. Accurate wind power predictions are essential to offer such services reliably. Over or underestimates
of energy and/or frequency control capability will result in economic loss either through curtailment and/or
penalties. The wind industry can facilitate forecasting by the development of further systems and meteorological



measurements at the wind turbine and plant level and by providing these measurements in real-time to weather
services and forecast providers (Lin and Liu, 2020).

One research area that still needs to be addressed is the coordination of plants in a region. Currently, when
frequency control services are required from wind plants, multiple wind plants respond. Similarly, as with SMs,
this can create an oscillatory effect if the collective response is too strong and not properly designed/tuned. Impact
of locational delivery of frequency response suggests that depending on where in the network frequency response
is injected, it could have a positive or negative impact (e.g., additional congestion). As wind power plants are
distributed, this can help in optimising the system wide coordination of delivery of frequency response (Wu et al.,

477    2018).


As offshore wind development will be combined with the development of offshore infrastructure based on HVDC
converters, supplying frequency control, especially the inertial response with grid forming control, will require
proper coordination of control with the multiple converters in the loop, i.e., wind turbine, offshore HVDC, onshore
HVDC, etc. (Gu et al., 2020). For HVDC connected offshore wind power plants, the controllers of the multiple
converters need to be properly coordinated (Glasdam et al., 2013; Sakamuri et al., 2017).
**3.2 Voltage control**
The effectiveness of voltage control is dependent on grid topology and location of the plant. Heavily loaded
transmission systems increase the risk for voltage instability. The main factors contributing to the long-term
voltage instability and subsequent voltage collapse are: i) stressed power systems with high active and reactive
power loading; ii) inadequate reactive power resources; and iii) load characteristics with respect to demand side
voltage (Van Cutsem and Vournas, 2007). The fluctuating nature of wind power can exacerbate the voltage
instability, playing a significant role in decreasing the dynamic voltage stability margin of the system (Zhou et
al., 2005).
Wind power plants can provide voltage control and have been doing so for many years. The impact of the
distributed nature of the wind resources on voltage services: if the wind resource is on a distribution grid, instead
of a transmission grid, can degrade its voltage control potential. Moreover, it is important to note that voltage
control is a localised need. While wind power plants can be used to enhance the voltage control in some locations,
there may be certain areas where the siting of wind turbines is not allowed.

Research is ongoing to better understand the limits and dynamic performance of reactive power provision from
wind power plants and how to use and coordinate them in practice (Qiao et al., 2009; Ghosh et al., 2020). Grid
forming operation of wind turbines will enhance their ability to provide voltage control, an intrinsic characteristic
of grid forming operation. Research is still needed on optimising the use of all power electronic devices installed
to gigawatt scale offshore wind farms (like Static Synchronous Compensator (STATCOM) installed at the onshore
connection point) to improve the dynamic range of reactive power capability. This reflects the highly integrated



nature of research in this area, i.e., the multitude of options to solve the challenges inside and/or outside the wind
plant (Veers et al., 2022).

Frequency and voltage control are services that wind is already providing typically as part of a portfolio of other
technologies such as SMs, and there are no specific long term research challenges for wind providing voltage
control.  As SMs disappear, the total reliance on wind and other IBR technologies will require additional
innovations and coordination with other services, particularly frequency control. The loss of inertial response will
require very fast frequency response using, e.g., grid forming controls which will also improve voltage control.
However, the proliferation of a multitude of IBRs in combination with reduced inertia, will drive the need for
services that address synchronisation and damping challenges, which are on the rise (Vittal et al., 2011) and are
dealt with in the next section.
**4. Synchronisation and Damping**
Frequency and voltage control, described above, is to maintain frequency and voltage within given ranges around
their nominal values. These ranges typically allow large deviations for short periods of time, sudden events, and
tighter ranges when the system is in steady state.  However, in steady state and during events there can be other
consequences that can be detrimental to the reliability of the power system, such as oscillations in the power
system that need to be damped and loss of synchronisation (Vittal et al., 2011; Wang and Blaabjerg, 2018).

**4.1 Synchronisation and angle stability**
As stated above SMs are at the heart of the synchronous AC power system that currently dominate worldwide.  If
SMs decline and are replaced by IBRs including wind, this synchronous characteristic starts to diminish, and a
loss of synchronism may occur which can result in the disconnection of SMs and hence a catastrophic loss of
generation and a subsequent blackout. To avoid a loss of synchronisation and instabilities induced by loss of
torque due to large load angles between groups of SMs, typically due to a large and abrupt supply demand
imbalance events, an increase in the active power output, proportional to rotor or voltage angle deviation is needed
(Boldea, 2005). For wind, the mechanism of loss of synchronism can be significantly different from SMs, and it
is highly dependent on the control method- grid forming or grid following. With more changes in the grid
following and grid forming wind, simple fault-ride-through is not enough and understanding the synchronisation
stability and the impact of power electronics (current-limiting) and control is a challenge (Denis et al., 2018).

Synchronisation will be more critical in the transition phase when there will be significantly less SMs to provide
this service. For power systems where there will be no SMs left, this synchronisation need will not exist.  There
may be a point in time in the transition, with very few SMs active in the system, when the challenge will be
temporarily very complex as the synchronous nature of the power system will give way to a non-synchronous
system. Hence this is a potentially transitory need and is all linked to the much bigger debate on the fundamental





nature of the power system going into the future and includes the debate around the ratio of grid forming to grid
following and needs and services with no SMs (Matevosyan et al., 2019; Bialek et al., 2021).

Synchronising service from wind is not provided today and is an emerging research area. IBR based wind power's
role in this challenge could be to mimic a SM, but this may not be the best approach and research effort should be
made to leverage the unique capabilities of the power electronics that interface the wind turbines to the grid (Pan
et al., 2020; Liu and Wang, 2021). The research challenge is mainly associated with the availability of the extra
active power needed during the re-synchronization period. The extra active power may also be needed
simultaneously for other services (frequency control, damping etc.) to meet other needs which makes the
coordination of the services from wind highly complex (Denholm et al., 2019). One solution could be to use some
of the overloading capabilities of wind turbines; however, this will depend on the duration and magnitude of the
disturbance and research is needed to understand the impact on expected performance and characteristics (Hansen
et al., 2014; Moawwad et al., 2014; Altin et al., 2018).  Another approach can involve the use of an energy source,
either directly connected to the wind turbine/plant or as part of a hybrid plant.
**4.2 Damping**
Power oscillations are generated because of interconnected power systems and power transfer operations. As
modern power systems continue to become more and more interconnected to provide adequate power and access
to capacity constraint-based corridors, the propagation of these oscillations is required to be tackled for a reliable
and secure power system operation. They can be induced by: (1) step changes in load, (2) sudden change of
generator output, (3) transmission line switching, and (4) short circuit, (5) change in operating point. In existing
power systems these power oscillations are caused by SM rotor angle swings varying from 0.1 to 4 Hz (Rafique
et al., 2022). Today, SMs are equipped with additional control loops, called power system stabilisers, which are
used to enhance the damping of power system oscillations through excitation control.

With the increased share of IBRs like wind (and solar PV), with their different characteristics – no "natural" inertia
– and dependence on control can lead to more frequent oscillations (Denholm et al., 2021). DFIG wind turbines
can cause sub synchronous oscillation (SSO) phenomena caused by the interactions between their controllers and
series capacitor compensators (Xu et al., 2019). Similarly, to synchronisation, the main challenge is the availability
of active power and the coordination with other services.

There are some examples of power system oscillations that are not fully understood and do not originate from any
clearly identifiable event but there is evidence that they are related to increased IBR penetrations, from wind and
solar PV, (Cheng et al., 2022). One of the biggest challenges is trying to understand their source, something that
also relates to the larger issue of having the correct models and tools to investigate these phenomena and to solve
them, typically by controller tuning (Miller et al., 2021).

There is a relatively large pool of literature related to power oscillation damping (POD) capabilities from wind
turbines and plants (Domínguez-García et al., 2012). The oscillations can be damped by injecting either active or



reactive power modulated at the terminal of wind plants (Zeni, 2015). It has been demonstrated through testing
that wind power can modulate their active and/or reactive power output based on different control algorithms to
provide damping services to the power system (Domínguez-García et al., 2012). It has also been demonstrated
that DFIG wind turbines operating in grid forming mode are less prone to SSO type instabilities due to the different
nature of their impedance characteristic (Shah and Gevorgian, 2020). While multiple POD control approaches for
wind turbines are presented in the literature, the capabilities have not been deployed or used in real life projects.
As stated above for frequency control and synchronisation, wind turbines are quite capable in modulating their
active or reactive power output. The challenge associated with that is mostly related to the availability of the extra
active power needed – if active power is chosen as the control variable. The coordination with other services is
also a challenge and hybrids with storage can alleviate this challenge and enhance this capability. A more
challenging issue is the possible impact of the active power modulation on the wind turbine drivetrain since the
low frequency oscillations typically occur inside (or very close) to its natural frequencies (Ghasemi et al., 2013).
Providing synchronising power and/or damping is not fundamentally a challenge to wind turbines. The main
research challenge will be related to the availability of the active power and, in the coordination of the control
with the other assets and services. While the power electronics may have some limitations, their controllability
gives additional capabilities that SMs cannot provide and may enable better operation of the power system
(Gonzalez-Longatt et al., 2021). These limitations are also a dominant theme of the final two power system needs
that are discussed below, protection and restoration, but the controllability may not bring significant advantage.
**5. Protection and Restoration**
Protection and restoration are extremely important needs for the power system. They protect human life and
technologies including generation, transmission, transformers from severe damage and against cascading failures
that may collapse the power system and/or in the event of a collapse allow the power system to be restored.
**5.1 Protection**
Over-current protection is currently a power system need and SMs are a source of large currents that will flow
during a fault. These currents are an essential triggering mechanism to activate the protection systems. The limited
overload-capabilities of power electronics in contrast to SMs, and hence the lower fault current capability of wind
turbines, can undermine the traditional way of triggering the protection system. The challenge with protection
also includes the fault current profile of IBRs including wind. It can vary between grid following and grid forming
control, and with the evolution of grid code requirement. The provision of negative-sequence current during
asymmetrical faults can also challenge the efficacy of relays. If SMs retire and are replaced by IBRs, these needs
for fault currents may not be met, and other protection methods may be required.
There are no known wind-based solutions for protection other than the fact that they can have some contribution
to the fault current. The fault current limitation of IBRs can be addressed by oversizing the power electronics, so



they can produce higher currents during faults. However, this solution is very costly, and will still not reach the
same response SMs can provide today (Bialek et al., 2021). DFIG wind turbines can provide much higher levels
of fault current (six times rated or higher) compared to full converter wind turbines (El-Naggar and Erlich, 2015).
DFIGs could therefore help the situation but would need to be deployed at many locations of the grid. Ways to
use the whole installed capacity of wind (and solar PV), that is much larger than SM capacity for providing the
same fault current demand could be explored. Wind power plants are most of the time generating less than rated
power and could run as STATCOMS when there is no wind available. However, the short circuit current will still
have to – individually – be limited to 1.5-2 times the rated current at each connection point. Potentially significant
changes are needed in protection methods once the SMs are all gone, pointing towards totally new approaches
like using travelling waves (Wilches-Bernal et al., 2021).

Making alternative wind turbine generator technologies available may address some of the shortcomings of
existing designs (Figure 2).  For example, Type V has been a concept for decades but is not commercially
available. It consists of a variable-speed drive system linked to a torque converter, operating in tandem with a SM
(Figure 6).  This SM can be seamlessly integrated into the grid through a circuit breaker, thereby conferring SM
benefits to the grid (Camm et al., 2009). It may provide many benefits including fault current contribution and
provision of inertia. These wind turbines are even capable of operating as synchronous condensers during periods
of low/no wind and may be a significant part of the future power systems (Henderson, 2021; Henderson and
Gevorgian, 2022).

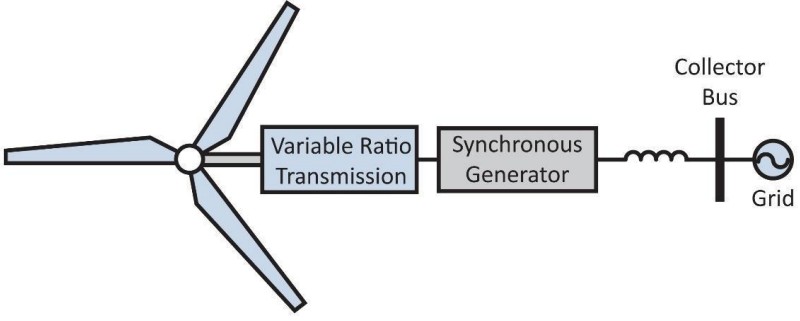





**Figure 6:  Type V wind turbine generator technology, not commercially available ((Osman et al., 2018), used with**
**permission).**

**5.2 Restoration**
Restoration of the power system following a blackout is an important need that is rarely activated. To start a power
system from a blackout situation requires a primary energy resource and the ability to form the grid (grid forming).



Wind with its power electronic interface can provide grid forming capability but can only provide an energy
source when the wind is blowing. Although this is a rather new topic for wind power plants, studies and first
demonstrations show that wind turbines equipped with power electronic interfaces have self-sustaining islanding
capabilities, meaning that – with a minimal energy source – they can start up, energise, and control their voltage
and frequency (ScottishPower, 2023). Preliminary field tests show that wind turbines can operate in grid forming
mode and energise a substation (Roscoe et al., 2020) illustrating that wind is capable of providing restoration
services when wind is available (Pagnani et al., 2020; ScottishPower, 2020). However, as a first stage of the
restoration process there are certain control and design modification challenges (Achilles, 2018). It could help the
power system if enough wind power plants are equipped with black start capabilities are located across the system,
in strategic locations considering network energisation and nearby loads including offshore (Jain et al., 2020).
Black start capable wind power allows implementing new bottom-up grid restoration strategies (as opposed to
conventional top-down approach by central thermal plants) helping to enhance grid resiliency and restoration
times.

As the restoration services from wind power is nascent, there are multiple aspects that need to be addressed that
from a needs and services perspective are like power system needs and services more broadly.  Individual wind
turbines can operate in grid forming mode, but when multiple wind turbines in a plant are jointly controlling the
voltage and frequency, often at quite low operation level (below 15-20% power), synchronisation and unwanted
interactions can occur (Lin et al., 2020; Henderson, 2023). Stability analysis methods and proper controller tuning
methods need to be developed. The wide timescale and frequency-coupling dynamics of power electronic
converters tend to bring in harmonic instability in the form of resonances or abnormal harmonics in a wide
frequency range (Wang and Blaabjerg, 2018), and the small signal stability of wind power plants operating in grid
forming mode for power system restoration (low power operating point) need to be investigated (Martínez-
Turégano et al., 2020). The chances of large transients occurring during restoration process are relatively high, as
the system will be in a vulnerable state, hence the performance of wind turbines operating in grid forming and
low power operating point needs to be investigated thoroughly (Papadopoulos and Milanović, 2016; Jain et al.,
2020). Hybrid with storage, wind and/or solar PV with grid forming are capable of self-black start first, then
energise transmission tie-lines, and participate in system restoration schemes (Bialek et al., 2021).

In times of blackouts, the provision of forecasts from external services may also break down. Power system
operators may have to work with older forecasts having worse quality.  Research and development on probabilistic
forecasting, providing reliable information about the available wind power is essential for restoration concepts
(NGESO, 2019).

Resilience, that is the ability of a power system to transition and potentially survive large scale events that may
otherwise cause a blackout and/or ensure the survivability of key assets that will allow restoration, is an evolving
area of interest to policy makers and the electricity industry more broadly (Panteli and Mancarella, 2015). It is
still in its formation stage as a power system need and is likely to either add to existing needs and/or result in new
needs and hence services.  For example, improving the resilience of power systems against extreme weather events





is one of the key concerns that designers and researchers are exploring, as the frequency of extreme weather events
is on a rising trend due to climate change (Mahzarnia, 2020). These events have the potential to cause severe
damage to power systems, leading to widespread blackouts that can have significant economic consequences (Bie
et al., 2017). The wind energy role in improving power system resilience involves research on turbines that can
withstand high-speed winds during hurricanes and in the restoration process (GE, 2018; Simpkins, 2022).

There are some common themes arising from protection and restoration that are also reflected across other
services, and this is discussed briefly in the final section along with some conclusions.

**6. Discussion and Conclusions**
Table 1 to 4 briefly summarised specifics around the status and opportunities, subject to research, for wind to
provide services in power systems.  They fall into two well defined groups.  Wind can and does provide Energy,
Capacity, Frequency and Voltage control services and these can be further improved and enhanced by research.
Wind does not currently provide Synchronisation, Damping, Protection and Restoration services but there are
nascent research efforts to investigate the potential of wind to provide these services and there is no fundamental
reason why they cannot provide them, however they may not be competitive nor groundbreaking and more
investigation is required.  This research needs to be done in the context of a fundamental change in power systems,
driven by the replacement of SMs with wind and other IBRs.  This transition is very dynamic and is presenting
profound new research challenges to the objective of maintaining supply demand balance reliably at least cost.
The simultaneous changes of increasing wind, solar PV, batteries, HVDC etc. and electrification - lead to a high
dimensional situation where the challenge is not a unidimensional "wind integration" challenge but a
multidimensional energy systems integration challenge.











**Table 1: Brief summary of wind's status and opportunity to provide energy and capacity services.**

| Service | Status | Challenge | Research and development needs |
|---|---|---|---|
| Energy and Capacity | Currently provided. | Availability of active power which is affected by weather and leads to variable output of wind power plants.<br><br>Low-capacity value (availability of wind at right timing).<br><br><br>A correct metric for resource adequacy in power systems dominated by wind and solar PV. | Forecasting: especially seasonal weather forecast accuracy.<br><br>Diversity to wind turbine technology: low wind turbines and long term potentially airborne wind power plants.<br><br>Long-duration storage in wind and hybrid power plants, including direct power-to-X application at wind plants. |













**Table 2: Brief summary of wind's status and opportunity to provide frequency and voltage control services.**

| Service | Status | Challenge | Research and development needs |
|---|---|---|---|
| Frequency control | Currently provided in some locations. | Availability of active power which is affected by weather and leads to variable output of wind power plants.<br><br>Coordinating various services with the required energy.<br><br>Revenue loss from Energy Curtailment for Frequency Services<br><br>Mechanical impacts on drive trains when procuring inertia-like response.<br><br>Potential frequency stability impacts. | Forecasting for frequency control services, including available power from wind power plants.<br><br>Grid forming operation of wind turbines for rapid frequency response.<br><br>Flow models and plant controls adequate for frequency control provision.<br><br>Methods and models for integrated simulation & analysis of mechanical & electrical wind turbine components<br><br>Wind-based hybrid systems for frequency support purposes.<br><br>Regional plant coordination for frequency stability |
| Voltage control | Currently provided in some locations. | Coordinating various services with the required energy. | Coordination of wind turbines to enhance voltage support capabilities. |





| | | Fluctuating nature of wind power. | Wind turbine coordination for enhanced voltage support.<br><br>Grid forming operation of wind turbines to provide voltage control.<br><br>Investigating the operational efficiency of power electronic devices within large-scale offshore wind farms. |
|---|---|---|---|



















**Table 3:  Brief summary of wind's status and opportunity to provide synchronisation and restoration**
**services.**

| Service | Status | Challenge | Research and development needs |
|---|---|---|---|
| Synchronisation | Currently not provided. | Availability of active power which is affected by weather and leads to variable output of wind power plants.<br><br>Coordinating various services with the required energy. | Wind turbine overloading capability for providing synchronisation.<br><br>Optimised operation of wind-based hybrid systems and battery storage connected to the wind turbine/plant. |
| Damping | Currently not provided. | Availability of active power which is affected by weather and leads to variable output of wind power plants.<br><br>Coordinating various services with the required energy.<br><br>Mechanical impacts on drive trains when procuring inertia-like response. | Power system oscillations, finding their source, and providing correction methods.<br><br>Optimised operation of wind-based hybrid systems and battery storage connected to the wind turbine/plant.<br><br>Implementing power oscillation damping in the field.<br><br>Mitigating mechanical impacts to drive train from active power modulation. |






**Table 4: Brief summary of wind's status and opportunity to provide protection and restoration services.**

| Service | Status | Challenge | Research and development needs |
|---|---|---|---|
| Protection | Can provide but maybe not to the extent that is necessary for existing protection schemes. Some wind technologies are better than others e.g. DFIG. | Restricted inverter overload current capacity.<br><br>Provision of negative-sequence current during asymmetrical faults. | Different wind technologies' capabilities to provide more fault current: e.g. type V turbines, hybrids, and larger fleet of wind/solar plants.<br><br>Alternative protection schemes that do not require overload current inverter capacity.<br><br>Specifying the fault current behaviour of grid-forming wind turbines when reaching the current limit. |
| Restoration | Currently not provided.<br><br>Capability being tested from grid forming turbines | Availability of active power which is affected by weather and leads to variable output of wind power plants.<br><br>Inverter overload capacity constraints. | Bottom-up grid restoration with the help of wind turbines.<br><br>Location of wind turbines that will be used to help restoration, needs to have small subsystems with |





| | | Small and large signal stability issues of grid-forming wind turbines in restoration scenarios. | equipment able to initiate restoration.<br><br>Hybrid power plants.<br><br>Weather forecast precision and role in the restoration process.<br><br>Small and large signal stability assessment of wind turbines in restoration scenarios.<br><br>Weather resilient wind turbines. |
|---|---|---|---|

Least cost solutions for resolving these research challenges can come from inside or outside the wind turbine/plant,
within the power system and/or from the broader energy system. Here the focus has been on solutions in the form
of services that come from within the wind technologies and those that are directly integrated with the wind
technologies e.g. power electronics and hybrids. This translates into a multiscale research and design challenge
at the individual turbine and plant level with mechanical, electrical and/or control centric solutions to the provision
of services (and/or reduction in needs) and a technical/economic comparison with other potential sources of
services to meet the power system needs.
The role that wind will play in the solution will be a measure of its competitiveness with respect to other solutions.
Resources, policy environment and stage of development are different across the world and there is and will be a
wide variety of power systems with distinct characteristics and while most power system needs and services will
share a lot in common some of them will be system specific. Therefore, wind technology in this rapidly changing
environment needs to adapt competitively leveraging its advantages and minimising its disadvantages on a
regional basis. For example, wind has the ability to respond more quickly and more accurately than a SM to meet
and provide some services and these may be an advantage depending on the power system dynamic characteristics
of the host power system.
The goal of the wind industry should now shift towards a more holistic minimisation of the portfolio cost while
ensuring higher value of wind power as well as maintaining and/or improving grid reliability. This holistic
approach means wind power needs to ensure coordination across its own technical capabilities and across multiple
time scales to maximise its value to the power system and minimise the cost to the wind technology. Maximising
the contribution of wind technology to the power system requires a deep understanding of the behaviour of the



wind technology with respect to its capabilities to provide services and how they interact and/or are dependent on
one another.  For example, wind power inherently has little stored energy and as many of the services require
energy then the provision of this energy is fundamental and acts to make the services highly dependent on one
another.  This coordination challenge is dominated by the provision of active power needed during service
provision as the active power may also be needed simultaneously for multiple services (e.g., frequency control,
damping, etc.) which makes the coordination of the services from wind highly complex. For example, the
interactions of wake steering, curtailment and the provision of services have not been fully investigated in detail.
Going beyond the turbine and plant the coordination/aggregation across a region brings challenges that are spatial
and temporal.  The temporal issue is not only in an operational time frame as the capital and maintenance costs
implications of providing services from wind technology are an important dimension of cost minimisation.
Since the earliest times in the industry forecasting has been a research challenge.  Coordinating services from
wind brings a new dimension to the forecasting challenge and as the energy system becomes more integrated the
coordination and forecasting across wind, solar, demand, new electrified demands and flexible demands makes
this an exciting area.  Its contribution is manifold, reducing needs e.g. reduced frequency control, increase in
reliability of wind service provision, and in a longer term the reliability of capacity services.
Not all the challenges within wind technology are directly coordination related.  For frequency control some open
issues remain for faster responses, like the ability of measuring, computing and transmitting the frequency signal
to the wind turbine/plant controller fast enough.  For best impact on capacity value, diversification of turbine types
is the key i.e. not just focussing on maximising energy capture but accounting for when the wind resource can be
converted into electricity by focussing on low wind speeds.  This diversification is also evident in offshore wind
and possibly future breakthroughs such as airborne wind and the ability of wind turbines to withstand extreme
weather.  As the penetration of IBRs increases there may be an inflection point where for example SMs disappear
completely from the power system and the need for synchronisation also disappears. However up to that point
when SMs are decreasing in number there may be a need to provide the synchronisation service from IBRs a
significant research challenge.  These "phase change" type transitions are in detail unknown and may form a
significant difference between power systems of the future where there may be several main types in contrast to
today where the synchronous power system is not the only and dominant type.  The opportunities for wind and
other technologies during these transitions are significant and potentially extremely challenging.
Wind technology interface to power systems and with most new devices is dominated by power electronics (IBRs)
and many of the potential evolving needs and services are being driven by the characteristics and nature of this
enabling technology. Power electronics differ significantly from SMs principally with respect to its characteristics
being determined by the control algorithms deployed within the physical limitations of the hardware in contrast
to SMs where the characteristics are driven almost entirely by the basic electromechanical physics of the hardware.
There are positives and negatives in this technology transition best summarised as positives emanating from the
controllability offered by the algorithms but with some physical limitation of the hardware e.g. over current
capability and the lack of direct mechanical coupling that can provide inertial response.

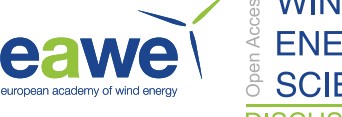

While the fundamental principles of IBR controls are a research topic regardless of the host technology, the
implementation in wind turbines will need to be adjusted or tailored to their characteristics. IBR based wind power
could mimic SMs, but this may not be the best approach as it may negate the controllability advantage. Research
effort should be focussed on leveraging the unique capabilities of the power electronics that interfaces wind to the
grid. There is a growing realisation that there is no one control approach that is most appropriate and that a mix
of control approaches will be best and that this will be system specific. This is best characterised by the so-called
grid forming or grid following approaches where for example grid forming mode would significantly decrease the
reaction time and, to some extent, mitigate for example the challenge with measuring, computing, and transmitting
frequency change.  But going all into grid forming is not recommended as it brings its own challenges.
Understanding the impacts and benefits of the different control approaches is a power system research focus but
the wind technology needs to be involved and participate so as the different control approaches can be designed
into the basic wind turbine technologies.  Not only should the wind industry work with the power systems
community but with the power electronics manufacturers to address these research challenges.
Many of the integration challenges can be solved without direct or indirect physical participation of wind
technology and as stated above this is very system dependent. While wind may not be the central part of a solution
it may play a part e.g. in protection while wind may not be capable of providing significant fault current it can
provide some and this can be part of a bigger solution. Regardless, wind as an integral part of future power systems
needs to provide information on its own performance and characteristics so as system operators can at a system
level determine and quantify the needs.  The control algorithms embedded in the power electronics that interface
to the power systems cause significant challenges to quantify the needs. They are vendor specific and proprietary;
they expand the degrees of freedom significantly at a device level and with the relatively small size relative to SM
resources they also expand the spatial dimensionality.  This is now causing real problems in many of the leading
power systems globally and therefore robust wind turbine/plant models that represent the power electronics and
its controls are urgently needed.  Allied to this challenge are the development of appropriate tools for wind power
plants and power system analysis (methods, component models and data) for both power system planning and
operations and wind technology needs to provide the detailed models and data to empower these tools and
methods.
The increasing share of wind power and other technologies is fundamentally altering the nature of power systems.
Wind power's role in future power systems will be driven by its ability to adapt to these changes by competitively
providing the required power system services.  This will require research and development that is done in
coordination with the other technological research communities including solar PV, power electronics and power
systems.  This is all driven by the unchanging nature of the fundamental objective of power systems - maintaining
supply demand balance reliably at least cost.






**Author contributions:** Mark O'Malley wrote the initial full draft of the paper - proposed the structure and conclusions, edited it extensively throughout the process and finalised the paper before submission. He did this with the assistance of Fatemeh Rajaei-Najafabadi and Andreas Hadjileonidas. Fatemeh Rajaei-Najafabadi was also the main contributor for the extensive referencing ensuring its accuracy and appropriateness at all stages.

Hannele Holttinen and Nicolaos Cutululis in collaboration with Mark O'Malley were the main architects of the structure and content, they also contributed significance to the technical details of many sections and were the main reviewers of the paper with respect to quality control.

Til Kristian Vrana, Jennifer King, Vahan Gevorgian and Xiongfei Wang reviewed the paper on several occasions - contributing to fine tuning several sections related to their own area of expertise and in general provided feedback of the paper.

**Competing interests**: At least one of the (co-)authors is a member of the editorial board of Wind Energy Science.

**Acknowledgements:** This article was written as an international research collaboration under IEA Wind TCP Task 25 "Design and Operation of Energy Systems with Large amounts of Variable Generation". It is part of the IEA Wind TCP Grand challenges set of articles, and the authors would like to acknowledge Paul Veers and Katherine Dykes for guidance. The authors would also like to acknowledge the review comments from Charlie Smith, Damian Flynn, Ana Estanqueiro, Jan Dobschinski, Niina Helistö, Matti Koivisto, Germàn Morales, Deepak Ramasubramanian, Tim Green, and Paul Veers.

**Financial support:** Mark O'Malley and Fatemeh Rajaei-Najafabadi were supported by the Leverhulme Trust International Professorship.

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
