# Peer review of "and services of the power system"

_Wind Energy Science, 2023_

## Referee Comment (RC1)

Grand Challenges i Wind Energy Science – Meeting the needs and services of the power system

General comment: the paper is well structured and a complete overview of the present status of wind plants services to the grid and their possible evolutions, challenges, and research needs. It is very useful not only to scientific community but to all wind energy stakeholders including policy makers.

| Principal criteria | Excellent (1) | Good (2) | Fair (3) | Poor (4) |
|---|---|---|---|---|
| **Scientific significance:**Does the manuscript represent a substantial contribution to scientific progress within the scope of of WES (substantial new concepts, ideas, methods, analyses, or data)? | X | | | |
| **Scientific quality:**Are the scientific approach and applied methods valid? Is sufficient information given so other researchers (in principle) can repeat the work? Are the results discussed in an appropriate and balanced way (consideration of related work, including appropriate references)? | X | | | |
| **Presentation quality:**Are the scientific results and conclusions presented in a clear, concise, and well-structured way (abstract conveys efficiently the essence of the paper; number and quality of figures/tables; appropriate, fluent, and precise use of English language)? | X | | | |

1. Does the paper address relevant scientific questions within the scope of WES? Yes, the paper presents an overview of the services that wind plants are or will be able to do to the grid and to the power system. This is a hot topic because wind energy is already well developed and a significant further growth is expected in the next decades with impacts on the power system.

2. Does the paper present novel concepts, ideas, tools, or data? Many concept are presented and broadly discussed, however the paper is not focused on one of them nor on presenting technical details.

3. Is the paper of broad international interest? Yes it is.

4. Are clear objectives and/or hypotheses put forward? Yes they are

5. Are the scientific methods valid and clear outlined to be reproduced? This is most an overview based on a high number of references

6. Are analyses and assumptions valid? Yes they are

7. Are the presented results sufficient to support the interpretations and associated discussion? Yes they are

8. Is the discussion relevant and backed up? Yes it is

9. Are accurate conclusions reached based on the presented results and discussion? Yes they are

10. Do the authors give proper credit to related and relevant work and clearly indicate their own original contribution? Yes they do

11. Does the title clearly reflect the contents of the paper and is it informative? Yes it does

12. Does the abstract provide a concise and complete summary, including quantitative results? Yes it does

13. Is the overall presentation well structured? Yes it is

14. Is the paper written concisely and to the point? It is not concisely because many concepts are presented, however there is the right space for each concept

15. Is the language fluent, precise, and grammatically correct? Yes it is

16. Are the figures and tables useful and all necessary? Yes they are

17. Are mathematical formulae, symbols, abbreviations, and units correctly defined and used according to the author guidelines? Yes they are

18. Should any parts of the paper (text, formulae, figures, tables) be clarified, reduced, combined, or eliminated? NO

19. Are the number and quality of references appropriate? Yes they are

20. Is the amount and quality of supplementary material appropriate and of added value? na

Suggestions:

line75 to what the "realiability" is referred?

Page 5 Figure 1 is not clearly readible

Line 270 to which "their" is referred?

Page 305 I don't seem to have found the definition of "capacity value"

Line 724 table 2 the second line of column "research and development needs" seems a rpetition of the first one

---

## Author Response (AR1)

**Comments on Grand challenges of Wind Energy Science**

RC1: 'Comment on wes-2023-179', Anonymous Referee #1, 07 Feb 2024
 The paper is well structured and a complete overview of the present status of wind plants services to the grid and their possible evolutions, challenges, and research needs. It is very useful not only to scientific community but to all wind energy stakeholders including policy makers. Very few editorial change are required.

Response: Thank you for your thorough review of the paper. We appreciate your positive feedback and we have carefully considered all your comments and addressed them as detailed below.  We have also made a few very minor changes to the English to make it more readable.

Suggestions

line75 to what the "realiability" is referred?

Response:  The text has been edited to clarify this, see line number 86 in the revised manuscript.

Page 5 Figure 1 is not clearly readable

Response:  Figure 1 now has only one figure not four and all the text is in the caption therefore it should be more readable.

Line 270 to which "their" is referred?

Response:  The text has been edited to clarify this, see line number 267 in the revised manuscript.

Page 305 I don't seem to have found the definition of "capacity value"

Response:  Capacity value is first used in the paragraph before this and the first reference used when referring to capacity value is Keane et al., (2011) which is an IEEE Task Force on the subject that defines it clearly.

Line 724 table 2 the second line of column "research and development needs" seems a repetition of the first one

Response:  We are not entirely certain about this suggesting but with the tables now spread out and not all in one place any confusion should now be avoided.
* * *
RC2: 'Comment on wes-2023-179', Anonymous Referee #2, 26 Feb 2024
The paper summarizes and discusses future challenges associated with the integration of wind turbines (and PV systems). Overall, the review is very comprehensive, although many aspects do not relate exclusively to challenges in the wind sector, but generally to the integration of renewable sources that are connected to the grid via power electronics. At first, I found it difficult to identify the actual content of the paper or the specific contribution. The structuring could be revised to some

extent and a stronger connection to the classification of stability definitions could also be established at this time. This could help the reader to better understand the chosen structure and the topics addressed. In addition, legal framework conditions, norms, and standards, which already require (or completely overlook) some aspects, should be addressed in some places. The role of politics and grid operators could also discussed more often.

Response:

We have split the introduction into two sections, Introduction (Section 1) and Integration of wind power in power systems (Section 2). At the end of Section 1 we have added a new paragraph to identify early on the content of the paper, see line number 73 to 81 in revised manuscript.

We assume that the reviewer is referring to the 2021 IEEE Transaction paper and other related work on classification of stability (Hatziargyriou et al., 2020). We have structured the paper around needs services as it captures the characteristics that are valued by system operators including stability. We have added some text to make this point, see line numbers 264 to 268 in the revised manuscript.

We were given the task by the leadership of the series to address technical research challenges. Legal framework, norms and standards, politics and grid operators are out of scope. There are other papers that address these issues and we have added some references and referred to them, see line number 75 to 78 in the revised manuscript.

Here are some specific questions to the authors and comments that could further improve the paper:

Abstract: The abstract of the paper gives the reader little idea of what the paper is actually about. What is the contribution? This should be better emphasized.

Response: We have rewritten the abstract to better reflect what the paper is about.

Line 49: How is a synchronous power system defined? Could you elaborate on this?

Response: The first sentence in footnote 4 is new in response to this comment.

Line 77: Please also provide a brief explanation of non-synchrounous. In the end, it's a question of control, isn't it?

Response: We have added footnote 7 to explain the difference.

Footnote 6: What about wind turbines that are equipped with a DFIG? The sentence about modern wind turbines doesn't really fit.

Response: We agree there is a subtly about DFIG turbines where only part of the power is converted to DC and then back to AC. We have added some text to the first sentence of footnote 6 to address this comment.

Figure1: Figure 1 appears very large and contains the main information in text form. I wonder whether four individual figures are really necessary.

Response:  We agree and there is now one figure rather than four in Figure 1 and the caption has been rewritten to replace the text that was in the figure.

Figure 2: The word "model" in relation to the different machine types is somewhat misleading.

Response:  We agree and have removed the word model in the Figure 2 caption.

Line 141: In which power and voltage range is type IV the most widespread? Does this apply to onshore and offshore turbines? And what exactly is meant by "now", i.e., type IV is now the most frequently installed type worldwide, if we compare all installed turbines?

Response:  Power and voltage range are not influential on the choice of wind turbine type. However, we do agree that this line is confusing and incomplete and have edited it, see line numbers 154 to 158 in the revised manuscript.

Line 163: What does it mean that the SMs may not be at the correct location? This paragraph is about frequency maintenance and not about voltage maintenance, which is a local problem. Please explain.

Response:  In steady state the frequency can be assumed to be the same throughout the power system but not during a fault.  The frequency is localised for the period of the transient and this is an integration challenge that is becoming more prevalent with increasing IBR penetration.  To clarify this, we have added a sentence and some new references in the subsection on frequency control, see line numbers 406 to 408 in the revised manuscript.

Line 277: The contribution should be introduced beforehand so that the reader knows exactly what the paper is about from the outset (see my comment on the abstract). Perhaps a closer connection to the well-known classification of stability definitions in the power grid would be helpful in understanding the structure of the paper.

Response:

This line has been superseded and removed by the response to your earlier comments, and the reference to stability definitions.  These responses ae repeated below for convenience.

We have split the introduction into two sections, Introduction (Section 1) and Integration of wind power in power systems (Section 2).  At the end of Section 1 we have added a new paragraph to identify early on the content of the paper, see line number 73 to 81 in revised manuscript.

We assume that the reviewer is referring to the 2021 IEEE Transaction paper and other related work on classification of stability (Hatziargyriou et al., 2020).  We have structured the paper around needs services as it captures the characteristics that are valued by system operators including stability.  We have added some text to make this point, see line numbers 264 to 268 in the revised manuscript.

Line 422, 439: Please give more details about the amount of stored energy that can be used to e.g., maintain the frequency.

Response:  The exact amount of kinetic energy stored in the wind turbine rotor will depend on the operational point and is hard to quantify. For the specific purpose of frequency control, it is estimated that around 5-10% of the rated power can be extracted from the blades without jeopardizing the stable operation of the wind turbine. To reflect this a sentence has been added, see line numbers 430 to 431 in the revised manuscript.

Line 557: The listed events are rather associated with oscillations in the low-frequency range. I miss a paragraph on harmonic stability, its possible root causes and possible solutions. This topic will be particularly interesting for future HVDC connections.

Response:  This is an excellent observation and highlights potential omission when using a needs and services paradigm and the evolving nature of the challenges.  Recognising this in general and specifically harmonics we have added a full paragraph with some additional references at the end of the Section 7, see line numbers 858 to 864 in the revised manuscript.

Line 695: I would be careful with such formulations.

Response:  We agree and have reworded this, see line numbers 764 to 767 in the revised manuscript.

Tables 1-4: I wonder if it wouldn't be better to place the tables at the end of each paragraph to summarize the (sub)section compactly. Please reconsider this. In this context, I would also appreciate if the tables contained references to literature that deals with the respective topic / challenge and discusses possible solutions.

Response:  We agree and have placed the tables at the end of each paragraph.  As the text with the detailed references are now adjacent, we have not put references into the tables.

Line 762: A general comment/question: to what extent can the win industry control this or should it come from standardization, grid operators or politics?

Response:  We have made some edits to the text and made this a recommendation, see line numbers 788 to 792 in the revised manuscript.

Line 835: Personally, I think the conclusion is too long. It could be divided into a discussion, recommendations and a "real conclusion".

Response:  We agree, and we have split the Discussion and Conclusion section into two sections, Section 7 Discussion and recommendations and Section 8 Conclusions.  We have also edited the conclusion to make it "real", see line numbers 783 to 785, 788 to 789, 832 to 834, 841 to 843, 852 to 857, 868 to 875 in the revised manuscript.

General remarks: There is often mentioned "wind and solar PV", even though the title (and journal) is clearly about wind. The focus should be more sharpened. Further, the figures generally do not have a high resolution. This should be improved. The text in the figures is also not so easy to read.

Response:

We think it is important to point out where wind and solar PV share some challenges as that will make the paper more useful. We specifically pointed this out on the first page of the paper with footnote 1, which has now been edited to make this point clearer. We have reviewed it and frankly speaking we cannot see any justification in removing the references to solar PV, if were to do so we would be missing an important opportunity to highlight one of the key recommendations of the paper – this is a holistic problem not just wind and needs to be dealt with holistically. We have also added a sentence in the conclusions highlighting the potential benefit of the paper to the Solar PV community, see footnote 1 and line numbers 874 to 875 in the revised manuscript.

We have replaced Figure 1 in response to a previous comment, we have also replaced Figure 4 with a higher quality colour version and Figure 2 has a better resolution than originally. The text should now be more readable.

\*\*\*\*\*\*\*\*\*\*\*\*\*\*\*\*\*\*\*\*\*\*\*\*\*

CC1: 'Comment on wes-2023-179', S M Shafiul Alam, 26 Feb 2024
Discussion on Type 5 Wind Turbine on page 20, lines 624 - 631 seeks improvement. Specifically, why is it not as broadly deployed as Type 3 or Type 4 and what activities are in progress to support Type 5 turbine's broader deployment? Helpful references are given below:

1) https://doi.org/10.1109/MELE.2021.3139246

2) https://iopscience.iop.org/article/10.1088/1742-6596/2626/1/012018/meta

3) https://doi.org/10.1109/PESGM52003.2023.10253195

4) https://doi.org/10.1109/PowerTech55446.2023.10202811

5) https://doi.org/10.1109/TPWRS.2019.2891962

Response: The reviewer has highlighted a very important omission on our part, we have failed to point out the drawbacks of Type V wind turbines which may go some way to explaining their lack of deployment. We have edited the text and added an additional reference, see line numbers 696 to 699 in the revised manuscript.

\*\*\*\*\*\*\*\*\*\*\*\*\*\*\*\*\*\*\*\*\*\*\*\*\*